# DNA Conformational Changes Induced by Its Interaction with Binuclear Platinum Complexes in Solution Indicate the Molecular Mechanism of Platinum Binding

**DOI:** 10.3390/polym14102044

**Published:** 2022-05-17

**Authors:** Nina Kasyanenko, Zhang Qiushi, Vladimir Bakulev, Petr Sokolov, Konstantin Yakovlev

**Affiliations:** 1Department of Molecular Biophysics and Polymer Physics, Saint Petersburg State University, 7/9 Universitetskaya emb., 199034 St. Petersburg, Russia; zqsdxx@163.com (Z.Q.); vbakulev@inbox.ru (V.B.); p.a.sokolov@spbu.ru (P.S.); 2Department of Analytical Chemistry, Saint Petersburg State Chemical-Pharmaceutical Academy, 14, Prof. Popov str., 197376 St. Petersburg, Russia; yakoniv@ya.ru

**Keywords:** DNA-platinum complexes, binuclear platinum (II) compounds, pyrazine, tetrazole, 1,3-propanediamine

## Abstract

Platinum anticancer drugs inhibit the division of cancer cells through a DNA binding mechanism. The bimetallic platinum compounds have a possibility for blocking DNA replication via the cross-linking of DNA functional groups at different distances. Many compounds with metals of the platinum group have been tested for possible antitumor activity. The main target of their biological action is a DNA molecule. A combined approach to the study of the interaction of DNA with biologically active compounds of this type is proposed. The capabilities of various methods (hydrodynamic, spectral, microscopy) in obtaining information on the type of binding of coordination compounds to DNA are compared. The analysis of DNA binding with platinum binuclear compounds containing pyrazine, tetrazole, 5- methyltetrazole, 3-propanediamine as bridging ligands in a solution was carried out with the methods of circular dichroism (CD), luminescent spectroscopy (LS), low gradient viscometry (LGV), flow birefringence (FB) and atomic force microscopy (AFM). The competitive binding of different platinum compounds to DNA and the analysis of platinum attachment to DNA after protonation of its nitrogen bases simply indicates the involvement of N7 guanine in binding. Fluorescent dye DAPI was also used to recognize the location of platinum compounds in DNA grooves. DNA conformational changes recorded by variations in persistent length, polyelectrolyte swelling, DNA secondary structure, and its stability clarify the molecular mechanism of the biological activity of platinum compounds.

## 1. Introduction

Since it is a well-known fact that nuclear DNA is the main target for platinum antitumor drugs in vivo, the study of DNA interaction with novel platinum compounds in vitro is an effective way to reveal the molecular basis of their possible biological activity. It is also the easiest way to select the most promising compounds for further testing. New trends in molecular medicine are associated with the creation and usage of multi-functional complexes to solve several problems at once [1,2,3,4,5,6,7]. In this connection, metal coordination compounds can act not only as anticancer drugs, but also as excellent agents that provide the involving of different ligands into multifunctional construction via coordination bonds. Note that binuclear coordination compounds can form bridging bonds between distant regions of biopolymers. DNA’s ability to form various ordered nanoscale structures by complexation with synthetic polymers, charged compounds, surfactants, and others agents may be used in these technologies as well. Therefore, the understanding of the molecular mechanism of DNA binding with coordination compounds is of great interest for many reasons.

Since the discovery of the anti-tumor activity of cisplatin (cis-DDP) [8,9], only a very few other platinum compounds were introduced into clinical practice [10,11,12]. It is known that the activity of platinum compounds may be improved by the introduction of heterocyclic ligands containing N donor atoms [13,14,15,16,17]. For example, tetrazole [18,19] and pyrazine [20,21] are used as ligands for coordination compounds. The usage of binuclear coordination compounds has high potential for drug design [19,22,23,24,25]. Unlike cisplatin, such complexes are usually electrolytes with good solubility in water. The understanding of how platinum compounds bind to DNA molecules in a solution may help to find their structural features responsible for preventing DNA replication. Indeed, the inhibition of cell division is usually explained by the coordination of platinum to DNA [26,27].

Water–salt solutions are close to the natural environment in the cell and are suitable to study the DNA interaction with potential drugs. Analysis of the DNA interaction with new compounds in vitro, combined with a comparison of the obtained data with the result of DNA-cisplatin binding, is a convenient method for selecting the most promising compounds for further in vivo testing.

It should be noted that a specific DNA methylation is one of the important events among the range of alterations found in tumor cells [28,29,30]. Therefore, during in vitro studies, we must keep in mind that similar variations of the chemical structure of DNA bases can influence DNA-platinum binding in vivo. In addition, cytosine methylation plays a crucial role in the development of acquired chemoresistance, including when using cis-DDP [31,32,33,34,35]. These events modify the electronic structure of heterocycles. Of course, this affects the binding of platinum to DNA. In this case, additional coordination of binuclear platinum compounds with DNA can promote the formation of complexes that prevent DNA replication.

DNA-platinum interaction has been studied in great detail, although there are still unresolved issues related to this interaction. Different methods have been used to consider this interaction. Some methods (NMR, spectral methods) provide information about binding positions and about the state of the secondary structure of DNA, other methods (electrophoresis, sedimentation, viscometry, light scattering) allow us to evaluate global conformational changes in DNA during the interaction, scanning microscopy makes it possible to visualize structural DNA changes in the formation of platinum-DNA adducts.

We should also mention such methods of analysis of DNA interaction with platinum compounds as the FRET-melting method and Fluorescent intercalator displacement (FID) assay. The FRET-melting method is one of the effective approaches for the analysis of the interaction of nucleic acids with different ligands, including platinum complexes with DNA quadruplexes [36]. It is known that platinum complexes have an affinity for G-quadruplex DNA [37,38]. G-quadruplexes formed in human telomeres are considered attractive targets for anticancer drugs. For example, it was shown that telomeres in cisplatin-treated HeLa cells are markedly shortened and degraded [39]. Possibly, a platinum affinity for G-quadruplex DNA and the role of N7 guanine in the binding of Pt complexes to DNA may explain the activity of cisplatin.

The fluorescence-based Förster Resonance Energy Transfer-melting method is based on the stabilization of nucleic acid structure induced by ligands. This method has been used, for example, to estimate whether a compound is a good quadruplex ligand or not [40,41]. We did not use this method in our work due to significant inconvenience when using high molecular weight DNA.

Fluorescent intercalator displacement (FID) is also one of the methods used to analyze the binding of ligands to DNA [36,42,43]. It is a convenient tool for understanding the type of binding and for assessing the relative binding affinities of compounds to DNA. A dye molecule with a greater fluorescence intensity, when bound to DNA, is used. It can be displaced by a ligand during its binding to DNA. As a result, one can see a reduction in the fluorescence intensity of the dye. The traditional FID is based on the intercalation of the dye molecule into DNA. Nevertheless, any mode of dye binding to DNA which may actually cause a decrease in fluorescence after dye release from DNA to solution can be used.

We propose an integrated approach to the analysis of DNA interaction with biologically active compounds, based on the usage of the set of experimental methods. Along with well-known methods, we propose to use such approaches as the analysis of DNA protonation in complexes with platinum compounds or the analysis of competition between various platinum compounds for binding sites on DNA, as well as competition between the selected platinum compound and the fluorescent dye DAPI, which can bind to DNA at known positions. We focused on the analysis of DNA conformational changes in complexes with binuclear platinum (II) compounds with tetrazole, 5-methiltetrazole, pyrazine, and 1,3-propanediamine ligands. The combination of spectral (UV spectroscopy, circular dichroism), hydrodynamic (low gradient viscosity and flow birefringence) methods, and atomic force microscopy (AFM) made it possible to monitor the state of the DNA secondary structure and change in the DNA conformation (persistent length, shape, size and polyelectrolyte swelling of the molecular coil). We have used 0.005 M NaCl as a supporting electrolyte. We examine the interaction of DNA with binuclear compounds containing one or two common heterocyclic ligands.

## 2. Materials and Methods

We have used the high molecular calf thymus DNA (Sigma–Aldrich, St. Louis, MO, USA). The molecular mass of DNA M = 10^7^ was determined by the usage of the value of the DNA intrinsic viscosity [*η*] (in dL/g) in 0.15 M NaCl with the formula [44]:$$\left[\eta \right]=6.9\cdot 10^{-4}\cdot M^{0.7}$$

DNA sample was dissolved in distilled water at a room temperature, and after 5 days of storage at 4 °C a certain amount of salt solution (1 M NaCl) was added to achieve 0.005 M NaCl. Then, this prepared DNA solution was filtered and used as the initial stock solution. The DNA concentration in a stock solution was determined from the difference in the absorbance ΔD at two wavelengths 270 and 290 nm after DNA hydrolysis at 100 °C in 6% HClO_4_ for 15 min [45].: This approach (determining the DNA concentration using pre-hydrolyzed solutions) makes it possible to control the stability of the DNA double helix in complexes by determining its molar extinction coefficient from the absorption of solutions at 260 nm: E_260_(P) = 31.1 × D_260_/*C*(DNA,%). If necessary, the DNA solutions in 1 M NaCl were prepared by the adding of NaCl solution with a higher concentration.

Figure 1 shows the structure of the platinum compounds used. Platinum compounds were obtained by procedures described in [15,16,17,23,24,25].

The binuclear **Pt1** compound has two common tetrazole ligands. Binuclear platinum compounds named as **Pt****2**, **Pt****3**, and **Pt4** (Figure 1) consist of two Pt(II) atoms with two ammonia ligands in cis-configuration and one bridging ligand (tetrazole, 5- methyltetrazole, or pyrazine). A sufficiently long chain (1, 3-propanediamine) in a trans-position to the chlorine ions links two platinum atoms in **Pt5**. We use also cis-DDP and trans-DDP compounds. DNA interaction with **Pt4**, trans- and cis-DDP were studied earlier [21,46,47,48]. Several experimental data for these compounds were obtained in this research for comparison.

### 2.1. Low Gradient Viscosity, (LGV)

The relative solution viscosity $\eta_{r}=\frac{\eta }{\eta_{0}}$ (where *η*—is the viscosity of the solution and *η*_0_—is the viscosity of the solvent) was measured at different velocity gradients g in the range of g = (0.5 ÷ 2) s^−1^. The usage of *η**_r_* value at *g*→0 and DNA concentration *c* gives the reduced viscosity of the DNA solution $\eta_{\mathrm{red}}=\frac{\eta_{r}-1}{c}$. The extrapolation of the dependence of $\frac{\eta_{r}-1}{c}$ on *c* to zero-concentration allows us to determine the DNA intrinsic viscosity [*η*]:$$\left[\eta \right]=\underset{c\to 0}{\lim}\left(\frac{\eta_{r}-1}{c}\right)$$

For DNA with molecular mass M > 2 × 10^6^ the model of swollen statistical coil in a solution can be used. The Kuhn’s model of polymer chain (freely jointed chain, FJC) with the hydrodynamic length *L* and the average length of the statistical segment *A* is suitable. The worm-like chain with the persistence length *p* can also be used to describe the conformation of DNA in a solution. The parameter <*h*^2^ > ^1/2^ (the mean square distance between the ends of the polymer chain) defines the linear size of the molecular coil. The relation <*h*^2^> = *LA* (for Kuhn’s model) is correct for macromolecules in the absence of volume effects (ideal solution). The length of the statistical segment *A* indicates the chain rigidity. For a long polymer chain with more than 10 segments the *A* value links with DNA persistent length *p* as *A* = 2*p*.

The value [*η*] for high molecular mass DNA is related to DNA conformation parameters by Flory’s formula (see, for example, [49]):(1)$$\left[\eta \right]=\frac{\Phi {\left(h^{2}\right)}^{\frac{3}{2}}}{M}=\Phi \frac{{\left(L2p\right)}^{\frac{3}{2}}}{M}\alpha^{3}$$

Here, Φ is the Flory parameter, M is the molecular mass of DNA, *L* is the hydrodynamic length of the DNA molecule, and α is the linear swelling coefficient defining the volume effects, including the polyelectrolyte swelling. For the real polymer solution we have <*h*^2^ > ^1/2^ = *α*(*LA*)^1/2^ = *α*(*L*2*p*)^1/2^.

In our research, we use viscometric titration. The dependence of the reduced viscosity of DNA solutions on the concentration of platinum compounds at constant DNA concentration reflects the change in the volume of DNA random coil at constant M and *L*. The swelling (or packing) of the molecular coil can be caused by variation in DNA persistent length *p* or/and by the change in the electrostatic interactions and polyelectrolyte volume effects (they contribute to α value).

### 2.2. Flow Birefringence (FB)

The birefringence Δ*n* of DNA solutions of different concentration value *c* was measured at different velocity gradients *g*. The value (Δ*n*/*g*) at *g* → 0 was determined. The laminar flow that occurs during the experiment provides the orientation of ellipsoidal molecular coils. The value (Δ*n*/*g*)*_g_*
*_→_* _0_/*cη*_0_ at *c* → 0 was used to find the dynamo-optical constant [*n*]. This value together with the intrinsic viscosity of DNA [*η*] can determine the optical anisotropy of the polymeric coil. The difference in the polarizabilities of the statistical segment (*α*_1_ − *α*_2_) along (*α*_1_) and across (*α*_2_) the axis of the DNA double helix can be calculated with the formula [50]:(2)$$\frac{\left[n\right]}{\left[\eta \right]}=\frac{4\pi }{45kTn_{s}}\frac{{\left(n_{s}^{2}-1\right)}^{2}}{n_{s}}\left(\alpha_{1}-\alpha_{2}\right)$$

The statistical segment of double-stranded DNA consists of 250–280 base pairs [39].

For DNA the following equation is valid:(3)$$\frac{\underset{g\to 0}{\lim}\Delta n/g}{\left(\eta_{r}-1\right)\eta_{0}}=\frac{4\pi }{45kTn_{s}}\frac{{\left(n_{s}^{2}-1\right)}^{2}}{n_{s}}\left(\alpha_{1}-\alpha_{2}\right)$$

Indeed, due to the giant intrinsic (inherent) optical anisotropy of DNA, which is two orders of magnitude greater than the optical anisotropy of the macromolecule due to the so-called shape effect, we can use the ratio obtained for certain concentrations of DNA without extrapolating to *c* = 0. Hence, an optical anisotropy of DNA statistical segment (*α*_1_ − *α*_2_) can be determined for any concentration of DNA from the value $\frac{{\left(\Delta n/g\right)}_{g\to 0}}{\left(\eta_{r}-1\right)\eta_{0}}$. It is known that
(*α*_1_ − *α*_2_) = *S*Δ*β* = (*A*/*l*)Δ*β*,(4)
where *S*—is the number of base pairs in the statistical segment, *l*—is the base pair length along the DNA axis (0.34 nm), *A* = 2*p* (*p*—is the persistent length of DNA), Δ*β*—is the optical anisotropy of a base pair along and across the axis of the DNA helix.

All hydrodynamic measurements in our research were performed at a temperature of 21 °C.

### 2.3. Spectral Methods

UV absorption spectra of DNA in complexes with platinum compounds were recorded using an SF-56 Spectrophotometer (OKB Spectr, Saint Petersburg, Russia). Circular dichroism (CD) spectra were recorded with Mark IV Autodichrograph (Jobin Ivon, Villeneuve-d’Ascq, France). The circular dichroism value ∆ε = ∆*D*/*cd* (∆ε is the difference in extinction coefficients determined from the difference in the absorption Δ*D* for left and right circular polarized light, *c*—is the DNA molar concentration, and *d* is the optical pathway). The luminescence of solutions was measured with a Hitachi-850 fluorescence spectrometer using a 1 cm–thick quartz cuvette after the solutions were held for 1 h at the ambient temperature. Luminescence excitation and the emission spectra were corrected for the spectral sensitivity of the instrument.

### 2.4. Atomic Force Microscopy (AFM)

The images of DNA and its complexes with platinum compounds were obtained on a freshly chipped mica surface. Appropriate exposure for 10 min was performed after applying a drop of a DNA solution with various compounds containing magnesium ions to better fix the DNA. Unfixed molecules were washed off. Then the sample was dried, and direct scanning was performed with NanoScope 4a (Bruker, Santa Barbara, CA, USA).

## 3. Results and Discussion

The absorption spectra and other experimental data for DNA-platinum complexes were usually obtained after one-day storage of resulting solutions at 4 °C. This was done to make sure that coordination bonds had formed.

Platinum compounds with heterocyclic ligands have an absorption in the UV region that partly overlaps with the DNA absorption band. In most cases, the contribution of platinum compounds to examined absorption spectra is much smaller than the absorption DNA. The linear dependence of the optical density of the solutions of platinum compounds on their concentration C(Pt) (in the used range of concentration) shows the absence of any aggregation in solutions with 0.005 M NaCl. A certain salt concentration is present in all investigated solutions to exclude changes in electrostatic interactions.

Assuming that the absorption of free binuclear compounds is the same as their absorption in complexes with DNA, we can calculate DNA spectra in complexes (see, for example, Figure 2A for **Pt2**). One can see that the binding of **Pt2** causes the emergence of a long-wave “shoulder” in the DNA absorption band with small hypochromism at low C(Pt) followed by a hyperchromism at higher C(Pt). Hereinafter, C(Pt) indicates the molar concentration of the compound. The same result was obtained for DNA-cis-DDP complexes (Figure 2B). The bathohromic shift of maximum can be caused by the involvement of N7 guanine (N7G) in binding. The increase in DNA absorption can be explained by the local destruction of DNA base pairs stacking. One can see similar spectral changes observed for DNA in complexes with **Pt1**, **Pt2**, **Pt3** and cis-DDP (Figure 3). The basic model for cis-DDP binding to DNA is the bidentate platinum coordination to adjacent guanines (or guanine and adenine) of the same DNA strand (so-called intrastrand crosslinks) [51,52,53]. It should be noted that cis-DDP can also form two coordination bonds with DNA bases from opposite strands (interstrand crosslinks) [54,55,56]. Interstrand crosslinks explain better the blocking of the DNA replication, although the number of these complexes in a cell is quite small. The binding like the chelate complex with N7 (N7G) and O6 of guanine can also be formed. It is known that the binding of positively charged ligands to N7G in a major groove of DNA induces the bathochromic shift of the band maximum with the hypochromic effect at 260 nm at low ligand concentrations followed by the hyperchromism at a high concentration of ligands in DNA solutions [57,58].

In this way, the similar changes in DNA spectrum in complexes with **Pt1**, **Pt2**, **Pt3**, and cis-DDP may specify that N7G is the important binding site for binuclear platinum compounds. From the comparison of the structure of these compounds follows that the difference in the structure and numbers of the common ligands does not affect the result of the experiment. We can assume that DNA interacts only with one platinum atom of binuclear compounds, and chromophores of platinum compounds (common ligands) do not participate in this binding.

Circular dichroism (CD) spectra of DNA-platinum adducts depend on the type of platinum binding to the macromolecule [59,60,61]. An increase in the intensity of the positive band in CD spectra observed for DNA adducts with cis-DDP at a low concentration of platinum compound indicates the formation of intrastrand crosslinks (Figure 4B). A decrease in the intensity of positive CD band registered at a higher cis-DDP concentration as well as for DNA-trans-DDP adducts indicates the other type of binding. The positive band amplitude in CD spectra of DNA in complexes with **Pt1**, **Pt2**, **Pt3**, **Pt4**, and **Pt5** (see, for example, Figure 4A) does not increase in contrast to that observed for DNA in complex with cis-DDP. We can explain this result by the absence of the appropriate type of platinum binding to DNA. The monodentate coordination of platinum atom to N7G may be supposed. It should be emphasized that the platinum compounds under investigation do not show themselves any optical activity.

Previously, we have studied in detail the interaction of **Pt4** binuclear compound with DNA in 0.005 M and 1 M NaCl [21]. Let us compare the influence of **Pt1**, **Pt2**, **Pt3** and **Pt4** on the DNA structure. All these binuclear compounds are electrolytes, unlike cis- and trans-DDP. They are soluble in water with the dissociation of chlorine ions. The positive charge of platinum compounds can increase after further replacing of chlorine ions by water molecules within the coordination sphere of **Pt2**, **Pt3**, **Pt4**. DNA charge density is extremely high, and the negative phosphates easily attach positively charged platinum complex ions. The electrostatic attraction of platinum compounds to DNA contributes to further coordination of platinum to heterocyclic nitrogen atoms of bases. N7 atoms of guanine (N7G) and adenine (N7A) are located in a more hydrophilic major groove of DNA. These sites are most accessible to formation of platinum adducts. The generally accepted viewpoint is that cis-DDP forms two coordination bonds with N atoms of the adjacent DNA purine bases (1,2 intrastrand bidentate coordination). The N7 and O6 atoms of guanine provide an attractive position for the location of positively charged complex ions, whereas NH_2_ of adenine is a less convenient neighbor. In this way, platinum binding with guanine is more favorable than a similar interaction with adenine. Other possible sites for the coordination of platinum (N3 cytosine and N1 adenine) are involved to the hydrogen bonds with complementary bases. They can be accessed only after a partial destruction of the double helix. Therefore, N7G or/and N7A atoms are first binding sites on the bases of double stranded DNA. The cis-DDP may form interstrand bifunctional adducts with the destruction of hydrogen bonds in the GC pair and some unwinding of the helix at the site of platinum binding. Monofunctional adducts can also occur. The addition of NaCl as a supporting electrolyte allows us to control the ionic strength in DNA solution. At high NaCl concentration (1 M) the polyelectrolyte swelling of DNA molecule is suppressed. The negative charges of phosphate groups are shielded effectively by Na^+^ ions. It was shown that cis- and trans-DDP do not interact with DNA in 1 M NaCl [46,47].

Similarly, DNA interaction with **Pt1**, **Pt2**, **Pt3** and **Pt5** is not observed at high salt concentrations. This result indicates the important role of electrostatic interactions in platinum binding to DNA. Contrary to that, it was shown that **Pt4** could form complexes with DNA in 1 M NaCl [21]. The charge of **Pt4** is 2+ after being dissolved in water. Previously it was shown that divalent and trivalent metal ions can interact with DNA in 1 M NaCl [58,62,63]. Apparently, a greater charge of **Pt4** compounds in DNA solution is responsible for the formation of their complexes with DNA in 1 M NaCl. The difference in binding of **Pt4** and other binuclear platinum compounds to DNA is observed also in 0.005 M NaCl (see Figure 4B). The compound **Pt4** has a heterocyclic pyrazine in contrast to a smaller ring of tetrasole and methyltetrasole in **Pt1**, **Pt2**, and **Pt3** compounds. We have suggested that the type of the heterocyclic ligands in the binuclear compounds determines the geometry of the compound and the ability of the coordination of both platinum atoms to DNA in bidentate complexes. Indeed, the binding of **Pt4** causes changes in DNA CD spectra similar to that observed for **Pt2** and **Pt3**, but at concentrations two times smaller (see Figure 4B). This can indicate that **Pt4** binds to DNA via two platinum atoms, in contrast to one platinum atom for **Pt2** and **Pt3**. Note that **Pt1** differs from other binuclear compounds by the presence of two common heterocyclic ligands. Nevertheless, **Pt1** causes a change in DNA absorption in complexes similar to that observed for DNA binding with **Pt2** and **Pt3**. This also indicates that heterocyclic ligands of **Pt1**, **Pt2**, and **Pt3** do not participate in the formation of complexes with DNA.

Global conformational changes of the DNA molecule in complexes with platinum compounds are observed by hydrodynamic methods (Figure 5). The combination of low gradient viscometry with flow birefringence method allows checking the volume of DNA macromolecular coil and its persistent length. Indeed, the segmental optical anisotropy (α_1_−α_2_) is determined by the number of the base pairs in DNA statistical segment (Kuhn segment). Since all platinum compounds have or acquire a positive charge in the solution, the change in DNA solution viscosity can also indicate the variation of DNA polyelectrolyte swelling. Figure 5A shows a fall in DNA solution viscosity with the rise of **Pt1**, **Pt2**, **Pt3**, cis-DDP and trans-DDP concentration (the experimental errors are indicated as a size of signs in Figures).

The decrease in reduced viscosity of DNA solution reflects the shrinkage of the DNA coil due to the excluded volume effects and/or the reduction in the chain rigidity (DNA persistent length). The dependences in Figure 5 indicate the existence of at least two binding modes for cis-DDP (at low and at high C(Pt)). There are bidentate complexes without destabilization of the double helix at low C(Pt) that cause a decrease in viscosity without hyperchromism and without change in the DNA optical anisotropy. Further decrease in DNA volume with the drop in DNA optical anisotropy (in DNA persistent length) with a local unwinding of the helix and some destabilization of DNA secondary structure is observed at C(Pt) > 3 ×10^−5^ M. Intrastrand crosslinks do not cause destabilization of the double helix because of the excellent steric coincidence of the distances between the adjacent bases in DNA strand and between the chloride ions in cis-DDP. We believe that the first adduct of DNA and platinum (1,2-intrastrand crosslink) is supplemented by the appearance of a second type of complexes transforming DNA secondary structure. The formation of interstrand crosslinks (coordination bonds of platinum with the bases on the opposite DNA strands) is becoming possible. Spectral data show an increase in DNA absorption with a decrease in the intensity of the CD positive band. Local bending of DNA helix can also induce a decrease in the optical anisotropy of DNA [21,48]. Trans-DDP binding also induces the decrease in the volume of the DNA molecular coil, but the increase in (α_1_ − α_2_) value is observed. The local destabilization in base stacking at high C(Pt) influences the (α_1_ − α_2_) value at higher trans-DDP concentration.

These experiments imply that trans-DDP and cis-DDP cause fundamentally different changes in DNA conformation. Indeed, their binding to DNA shows different effects on DNA optical anisotropy, on DNA volume in a solution, and on DNA spectral properties.

The decrease in the reduced viscosity of DNA solutions and change in DNA segmental optical anisotropy as a result of DNA binding with **Pt1**, **Pt2**, and **Pt3** show that these binuclear compounds have a similar influence on DNA conformation to that observed for cis-DDP–DNA interaction (Figure 5A,B). However, in this case, we cannot distinguish two types of binding from the data of hydrodynamic methods, although the spectral data indicate this (Figure 2).

Figure 5 shows also that the binding of the compound **Pt5** (a binuclear compound with a long linker in trans-configuration) induces an increase in the DNA intrinsic viscosity in contrast to all binuclear platinum compounds under study, because of intermolecular crosslinking (Figure 5C). Both platinum atoms in **Pt5** link non-adjacent DNA groups. The DNA optical anisotropy changes with the rise of **Pt5** concentration similar to that observed for complexes with **Pt1**, **Pt2** and **Pt3**, but at a concentration twice smaller (similar to **Pt4**), which may indicate the formation of DNA bonds with two platinum atoms. Figure 5C indicates also the unique type of **Pt4** binding to DNA. The compound structure allows DNA binding with two platinum atoms. The drop of viscosity and of DNA optical anisotropy at low **Pt4** concentration is followed by their increase at higher C(Pt4). That can be explained by two alternative binding modes in DNA solutions at different concentrations of **Pt4**. The structure of the compound provokes a sharp bend of DNA double helix (a kink) after the formation of two coordination bonds of **Pt4** with DNA. This bending causes change in the DNA optical anisotropy (in the persistent length of DNA). The drop in the persistent length of DNA causes a corresponding decrease in DNA intrinsic viscosity. With the rise of C(Pt4) the second type of binding (DNA binding with one platinum atom) becomes more profitable, and the DNA conformation transforms similar to that in complexes with trans-DDP. The pyrazine ligand can be fixed in the major groove of DNA.

We emphasize once again that the experimental data allow us to conclude that **Pt1**, **Pt2**, and **Pt3** demonstrate similar influence on DNA conformation and have common features with that of cis-DDP-DNA complexes, with the exception of the ability to form crosslinks. We eliminate the bidentate coordination of each platinum atom in **Pt1**, **Pt2**, and **Pt3**. We suggest that DNA interacts only with one platinum atom of these compounds.

Fluorescent intercalator displacement (FID) is a convenient tool for understanding the type of binding and for assessing the relative binding affinities of compounds to DNA. We have used the fluorescence dye DAPI and analyzed its possible competition with platinum complexes for the binding sites in a minor groove of DNA. It was done to clarify the position of platinum complexes on DNA. It should be noted that DAPI is not a classic intercalator, in contrast to EtBr, for example. However, the method used in this work is based on a similar approach.

The luminescence of DAPI (4′,6-diamidino-2-phenylindole) and EtBr (ethidium bromide) dyes after their binding to DNA-platinum adducts was studied to elucidate the binding sites of biplatinum compounds on DNA [64,65,66]. It was shown that DAPI can form complexes in the DNA minor groove with the preferential binding to A-T pairs [67,68]. The possibility of dye intercalation or partial intercalation with the formation of hydrogen bonds with DNA groups in the minor groove is being discussed. When DAPI heterocycles intercalate (or partially intercalate) between planar hydrophobic DNA bases, DAPI becomes virtually unsusceptible to external influences and demonstrates a very high fluorescence quantum yield. This binding with the binding constant K = (3.0 ± 0.5) × 10^6^ M^−1^ and with the number of binding sites *n* = 0.02 (approximately one DAPI molecule per 50 DNA base pairs) causes the DAPI luminescence with the maximum at 460 nm (λ_ex_ = 340 nm). An increase in DAPI concentration causes its binding with DNA phosphates with the DAPI luminescence at 540 nm (λ_ex_ = 420 nm). When the concentration ratio C[DAPI]/C[DNA] reaches *z* = 0.3 two binding modes can be well distinguished in the luminescence spectra (Figure 6). We studied the formation of DAPI binding to DNA after or before platinum coordination to DNA. Figure 6 demonstrates data obtained for DNA complexes with **Pt2**. Other binuclear compounds show similar results.

Two different schemes were developed for the mixing of DNA, DAPI, and **Pt2** solutions in 0.005 M NaCl: (1) DNA-DAPI complexes were formed, and **Pt2** solution was added later; (2) the DAPI solution was added to the solution with DNA-**Pt2** complexes (after one-day storage of DNA solution with **Pt2**). The measurements were performed one day after the second mixing. The concentrations of DNA, DAPI, and **Pt2** were the same in all systems including the control DNA-DAPI and DNA-**Pt2** solutions. Cis-DDP was used for comparison in a similar experiment.

The absorption band of DAPI (out of DNA absorption area) in complexes with DNA is not noticeably changed after the addition of cis-DDP. Indeed, the dye is localized in the minor groove of DNA or near the phosphates, whereas cis-DDP interacts with N7G or N7A in the major groove. On the contrary, the addition of **Pt2** returns DAPI absorption to that typical for the free DAPI molecules. One can see this result in all solutions with DNA-**Pt2** complexes. We can conclude that the presence of **Pt2** in a solution prevents DAPI from binding to DNA.

The DAPI luminescence in complexes with DNA at λ_ex_ = 340 nm (the maximum of the DAPI absorption band) at *z* = 0.3 produces a wide band c with a maximum at 490 nm due to the coexistence of several spectral forms of DAPI (Figure 6B). Two different DAPI-DNA complexes (a strong binding in the minor groove and an electrostatic binding with the DNA phosphates) and free DAPI determine the luminescence of examined solutions. The addition of cis-DDP to the solution with DNA-DAPI complexes, as well as DAPI binding to DNA-cis-DDP complexes (Figure 6B, spectra 2 and 4) cause a slight decrease in the DAPI luminescence without sufficient change in the shape of the spectrum. This result shows that cis-DDP does not prevent the binding of DAPI with DNA in a minor groove. The addition of **Pt2** to the solution with DNA-DAPI complexes almost completely quenches the luminescence of DAPI associated with phosphates, but has no influence on the strong dye binding in the DNA minor groove. We can conclude that the electrostatic interaction of binuclear compounds with DNA phosphates plays a more visible role compared to that of cis-DDP. It follows that **Pt2** is located near phosphates or within the major groove and does not prevent DAPI binding with DNA in the minor groove. One can see that in this experiment the addition of **Pt2** really quenches the luminescence of DAPI localized on DNA phosphates.

The main conclusion from these experiments is that **Pt2** and cis-DDP are located out of DNA minor groove. However, **Pt2** binding to DNA more effectively prevents the DAPI interaction with DNA phosphates.

How can we check whether the coordination bonds of DNA with binuclear compounds are formed? It is known that during the equilibrium binding of ligands with DNA the ratio of free and bound ligands in a solution change with the varying in DNA concentration. It is necessary to keep this ratio constant with the dilution of the stock solution during the experiments. It is known that the intrinsic viscosity of DNA can be determined by the extrapolation of the dependence of reduced viscosity on DNA concentration c(DNA) to c(DNA) = 0. This dependence becomes nonlinear when the ratio of free and bound ligands changes. However, for complexes with coordination of platinum to DNA (being formed for quite a long time—i.e., few hours) the intentional change in the concentration of free ligands does not has time to influence platinum—DNA binding during the experiment.

We prepare the stock solution with DNA-platinum complexes. The dilution was implemented in two different ways. In the first case, 0.005 M NaCl was used for the dilution (the ratio C(Pt)/c (DNA) was constant in the experiment). The second dilution with constant C(Pt) in DNA solutions was also performed (the solution of a platinum compound with initial concentration was used as a solvent). The duration of the experiment (within 1 h) allows us to exclude the formation of the additional coordination bonds of platinum to DNA. These two dilutions change the balance between fractions of free and bonded compounds. In our experiments, both methods of dilution lead to similar concentration dependences for DNA complexes with cis-DDP and **Pt5** (Figure 7A). The results of similar experiments with **Pt1**, **Pt2**, and **Pt3** (Figure 7B) show that the correct extrapolation of the dependences to c(DNA) = 0 is impossible. Hence the reason why we cannot determine the intrinsic viscosity of DNA in complexes with compounds **Pt1**, **Pt2**, and **Pt3**. But we must remember an important role of electrostatic interactions with phosphates for binuclear platinum compounds in addition to the experimentally observed strong binding of platinum to DNA bases. This binding with phosphates certainly depends on the dilution method. In this case, the fraction of molecules with platinum coordinated to DNA bases remains unchanged, but other binding with phosphates noticeably changes. This result indicates the predominance of the equilibrium binding of platinum compounds with DNA phosphates, but does not exclude the formation of a small number of coordination bonds.

Another way to test the formation of the platinum coordination bonds with N7 of guanine is to study the protonation of DNA molecules in complexes with platinum compounds [48]. The absorption and the CD spectra of the protonated DNA [48,69,70] reflect the emergence of positive charges on the DNA bases. It is known that the primary proton acceptor group for double-stranded DNA is N7G. Indeed, other suitable groups N1A and N3C are involved in hydrogen bonds between complementary strands. DNA protonation depends on NaCl concentration in a solution (pK = 4.75 in 0.005 M NaCl and pK = 3.1 in 1 M NaCl). A typical change in the CD spectrum of DNA as a result of protonation in 0.005 M NaCl is shown in Figure 8A. We do not observe the DNA protonation after the formation of its complexes with cis-DDP, trans-DDP, **Pt4**, and **Pt5** in 0,005 M NaCl at pH < pKa (Figure 8). Our experiments indicate that platinum atoms in **Pt4**, **Pt5**, trans- DDP and cis-DDP compounds coordinate to N7G and block this position for DNA protonation. The absence of **Pt5** coordination to N7G in 1 M NaCl is evident from the spectra (Figure 8E) demonstrating DNA protonation under these conditions (note that for double-stranded DNA in 1 M NaCl the pK value is smaller than in 0.005 M NaCl).

We also study the competition for the binding site on DNA between several coordination compounds. Figure 9 shows that **Pt4** binding prevails over the interaction of cis-DDP with DNA. The binding with N7G occurs for both compounds. It was shown that **Pt4** coordinates to DNA via two platinum atoms and induces bending of the double helix. Our results indicate that the binding of **Pt4** blocks the formation of bidentate complexes of cis-DDP with DNA. The competition for the binding sites on DNA between **Pt1**, **Pt2**, and **Pt3** with cis-DDP is not so pronounced. However, the initial binding dominates. Note that in this experiment the concentrations of compounds are not high enough to completely block the potential binding sites on DNA bases.

Finally, let us compare the AFM images of DNA complexes with platinum compounds (Figure 10). For DNA complexes with cis-DDP one can see the occurrence of DNA shrinkage at C(cis-DDP) = 4 × 10^−5^ M, and a lot of kinks in the DNA helix at C(cis-DDP) = 5.5 × 10^−5^ M. DNA complexes with trans-DDP cause a shortening and thickening of the DNA strands with the occurrence of intra- and interstrands crosslinking. The local destabilization of the double helix [64] allows us to explain the observed structures. The trend towards shrinkage of the molecular coil and further curtailing the chain is also observed for DNA complexes with binuclear platinum compounds.

## 4. Conclusions

The experimental data obtained by different methods demonstrate a good agreement, and point to the main reasons for the observed change in DNA conformational parameters. Our experiments show that all binuclear platinum compounds interact with DNA in a solution with the predominant binding with phosphates at the first stage of interaction. The coordination of one platinum atom in **Pt1**, **Pt2**, and **Pt3** to N7G is also clearly observed in the experiment. The binding of those binuclear platinum compounds does not overlap the minor groove of DNA—the binuclear molecules are located near the phosphates and in the major groove. **Pt4** due to its geometry binds to DNA through two platinum atoms. This binding induces great bending of the double helix. Each platinum atom in **Pt5** can coordinate via monodentate complex to separate sites on DNA molecule due to a long linker chain. The proposed experimental approach gives information on the changes in the secondary and tertiary DNA structure as a result of DNA-platinum interaction. It can clarify the mode of platinum binding to DNA. Indeed, the combination of spectral and hydrodynamic methods with AFM using different concentrations of platinum compounds in a DNA solution shows different ways of their binding with different influences on the DNA conformational parameters. From the comparison of DNA binding with different platinum compounds, one can offer the correct models of interaction.

## Figures and Tables

**Figure 1 polymers-14-02044-f001:**
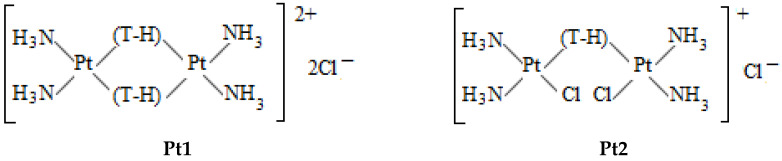
The structure of binuclear platinum compounds.

**Figure 2 polymers-14-02044-f002:**
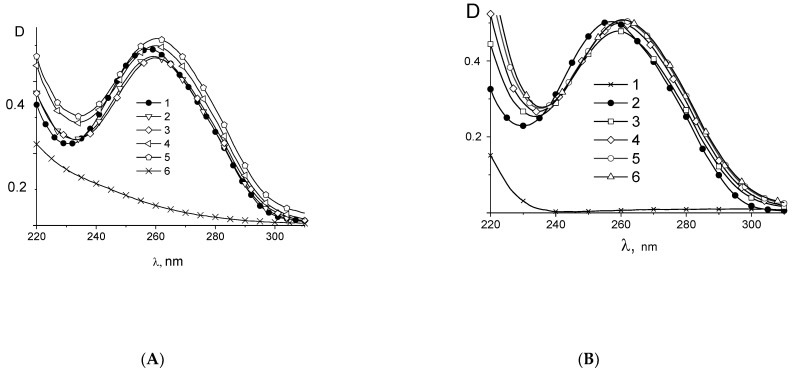
(**A**) Calculated DNA absorption spectra in complexes with **Pt2** in 0.005 M NaCl (**A**) at C(Pt2) × 10^5^ = 0 (1), 1.5 (2), 3.4 (3), 4.9 (4), 6.4 (5) and spectrum of Pt2 without DNA at C(Pt2) = 3.42 × 10^−5^ (6), C(DNA) = 0.0024% =7.2 × 10^−5^ M (P); (**B**) absorption spectra of DNA in DNA-cis-DDP complexes C(Pt) × 10^5^ = 0 (2), 1.2 (3). 2.2 (4), 3.9 (5), 5.6 (6). Spectrum for cis-DDP (1) was registered at C(Pt) =3.9 × 10^−5^ M.

**Figure 3 polymers-14-02044-f003:**
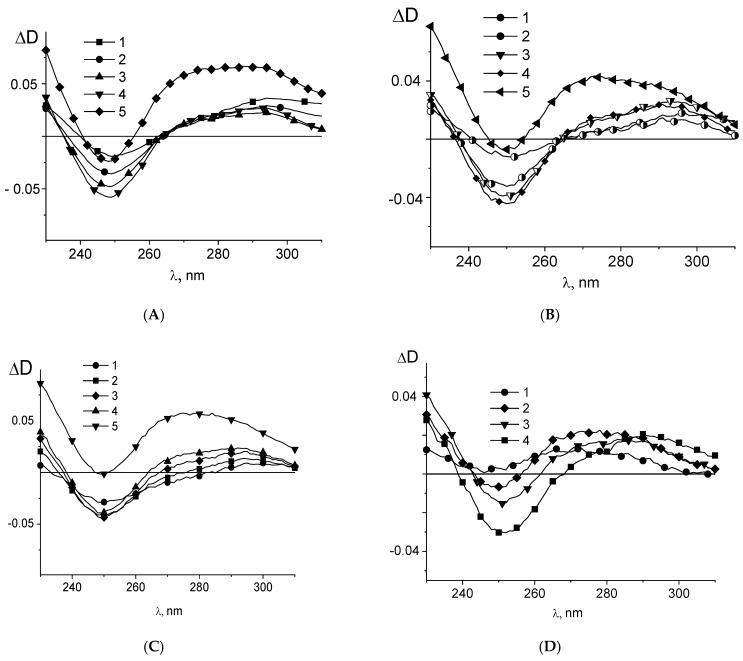
Differential absorption spectra (calculated absorption of DNA in complexes minus absorption of free DNA) for DNA complexes in 0,005 M NaCl with cis-DDP (**A**) at C(Pt) × 10^5^ = complexes C(Pt) × 10^5^ = 1.2 (1). 2.2 (2), 3.9 (3), 4.1 (4), 5.6 (5); for DNA complexes with **Pt2** (**B**) at C(Pt2) × 10^5^ = 1.5 (1), 2.0 (2), 3.9 (3), 4.5 (4), 6.4 (5); for DNA complexes with Pt3 (**C**) at C(Pt3) × 10^5^ = 1.6 (1), 2.4 (2), 3.8 (3), 4.6 (4), 6.4 (5) and for DNA complexes with **Pt1** (**D**) at C(Pt1) × 10^5^ = 0.54 (1), 2.7 (2), 3.8 (3), 5.4 (4).

**Figure 4 polymers-14-02044-f004:**
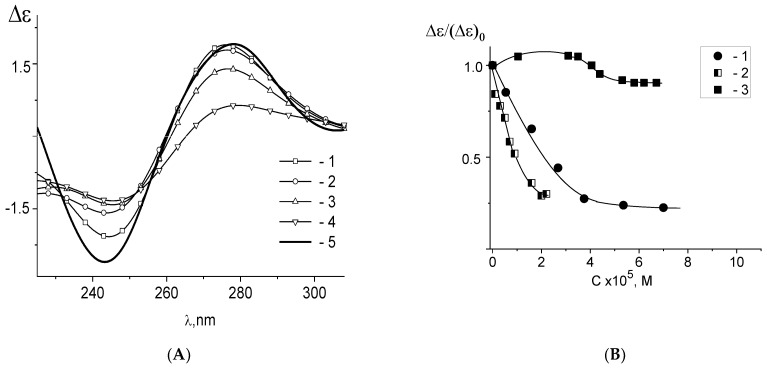
CD spectra of DNA in complexes with **Pt3** in 0.005 M NaCl (**A**), and the dependence of the relative change in the amplitude of positive maximum of DNA CD spectra (**B**) on concentrations of **Pt3** (1), **Pt4** (2) and cis-DDP (3).

**Figure 5 polymers-14-02044-f005:**
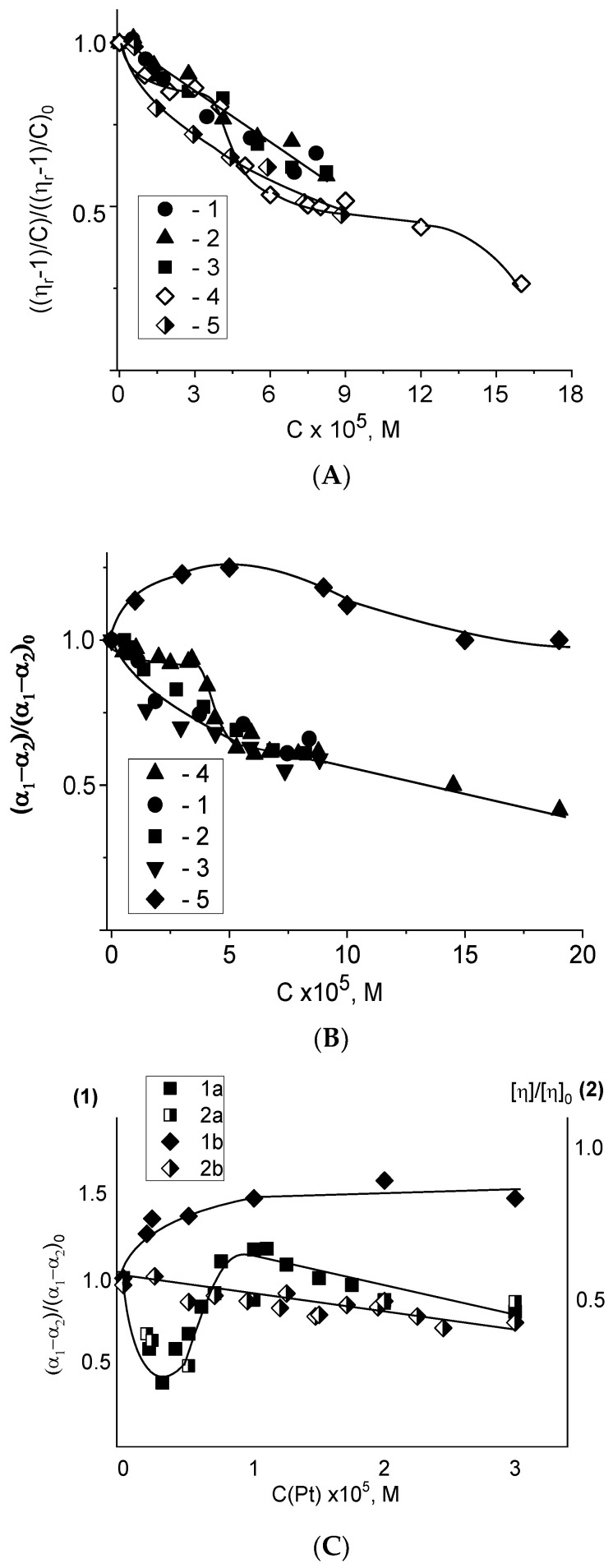
Relative change in the reduced viscosity of DNA solutions (**A**); in DNA segmental optical anisotropy (**B**) with the increase of C(Pt) for **Pt1** (1), **Pt2** (2), **Pt3** (3), cis-DDP (4), trans-DDP (5). (**C**) shows the dependence of the relative change in the segmental optical anisotropy (1) and in the intrinsic viscosity (2) of DNA on C(Pt) for **Pt4** (a) and **Pt5** (b). All measurements were carried out in 0.005 M NaCl.

**Figure 6 polymers-14-02044-f006:**
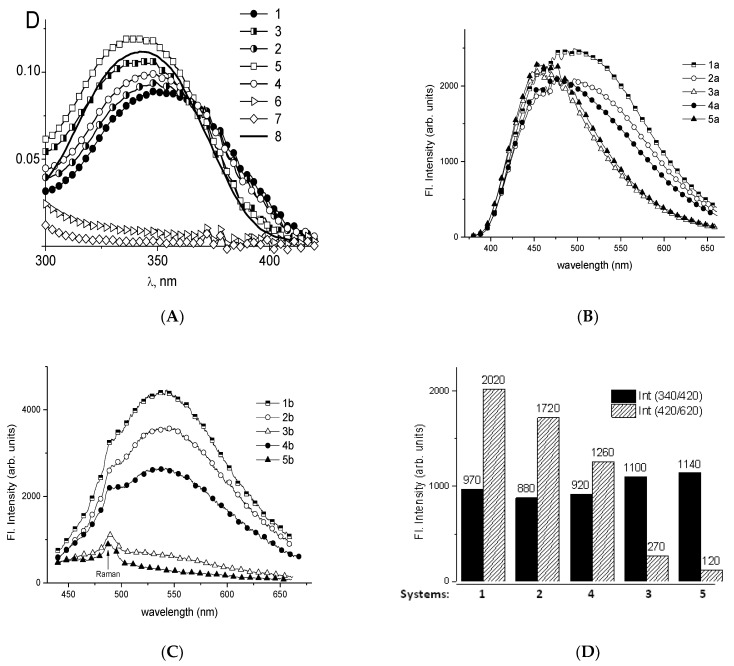
Absorption spectra (**A**) of DAPI in complexes with DNA in 0.005 M NaCl (1), DAPI in complexes with DNA after the addition of cis-DDP (2) and after the addition of **Pt2** (3); after the addition of DAPI to the solution with DNA-cis-DDP (4) and with DNA-**Pt2** (5) adducts; absorption of DNA-**Pt2** (6), DNA-cis-DDP (7) and DAPI (8) solutions. C(DNA) = 1.5 × 10^−5^ M, C(DAPI) = 5 × 10^−6^ M, C(Pt) = 4 × 10^−5^ M. Luminescence spectra (**B**,**C**) of DAPI in complexes with DNA in 0.005 M NaCl at λ_ex_ = 340 nm (a) and λ_ex_ = 420 nm (b) for same systems. (**D**) Comparison of data presented in (**B**) and in (**C**): black bars—the strong binding of DAPI in DNA minor groove (λ_ex_ = 340 nm, λ_em_ = 420 nm); shaded bars—the electrostatic binding of DAPI with DNA phosphates (λ_ex_ = 420 nm, λ_em_ = 620 nm).

**Figure 7 polymers-14-02044-f007:**
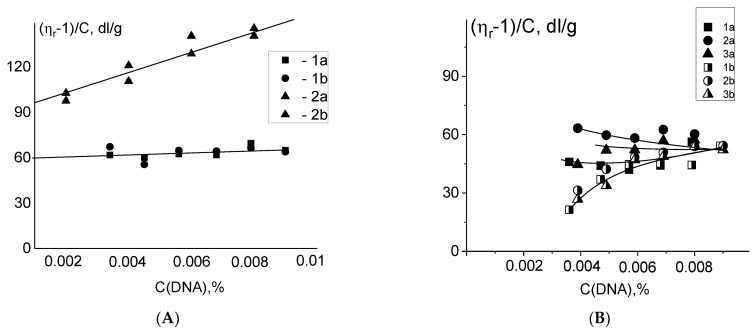
Dependences of the reduced viscosity of solutions on DNA concentration. Two methods of dilution of a stock solution with DNA-platinum complexes were used: at C(Pt)/C(DNA) = constant (a) and at C(Pt) = constant (b). (**A**) Results for DNA-cis-DDP adducs (1) and DNA-Pt5 adducts (2) are shown; (**B**) results for DNA adducts with **Pt1** (1), **Pt2** (2) and **Pt3** (3) in 0.005 M NaCl are shown. C(Pt1) = C(Pt2) = C(Pt3) = 6 × 10^−5^ M.

**Figure 8 polymers-14-02044-f008:**
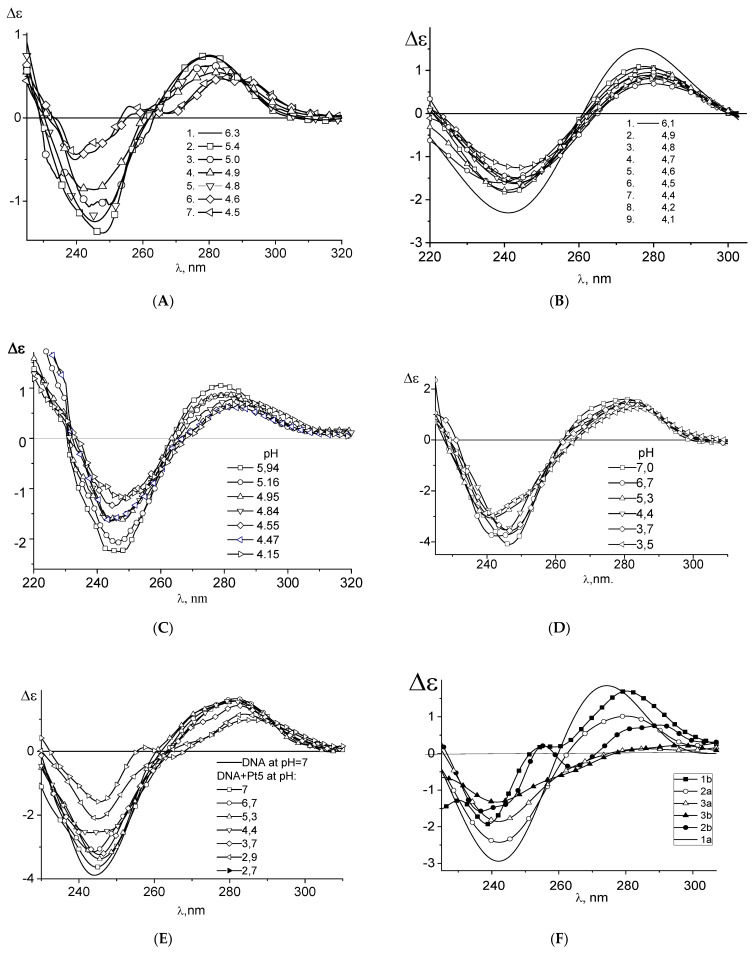
Circular dichroism spectra of DNA in 0.005 M NaCl (**A**) and of DNA complexes in 0.005 M NaCl with cis-DDP at C(cis-DDP) = 2.8 × 10^−6^ M (**B**); with **Pt4** at C(Pt4) = 6 × 10^−6^ M (**C**); with **Pt5** in 0.005 M NaCl at C(Pt5) = 5 × 10^−6^ M (**D**); with **Pt5** in 1 M NaCl at C(Pt5) = 5 × 10^−6^ M (**E**), pH values are given near the lines.; and with **Pt3** (**F**) in 0.005 M NaCl at C (Pt3) = 0 (1) 6 × 10^−6^ M (2) and 1.07 × 10^−5^ M (3), at pH = 6.2 (a) and 4.2 (b).

**Figure 9 polymers-14-02044-f009:**
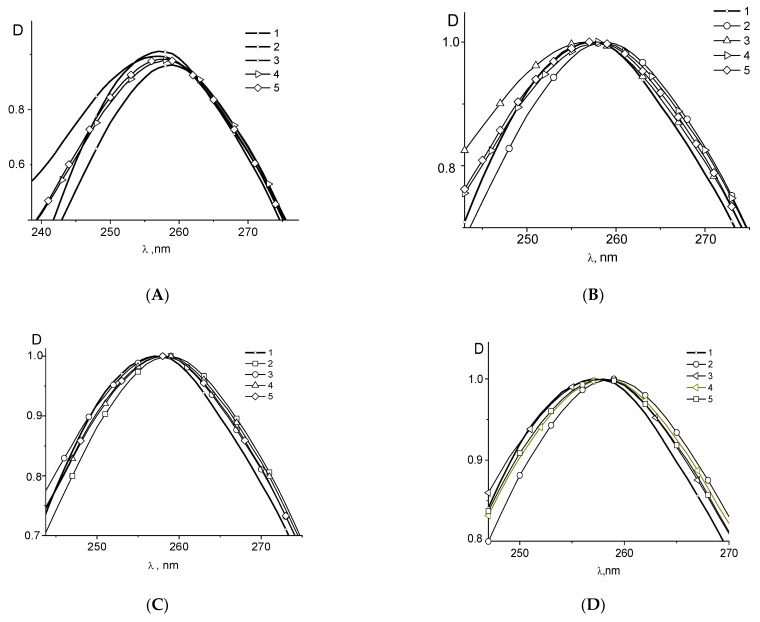
UV (**A**–**D**) and CD (**E**) DNA spectra (1) and normalized DNA spectra in complexes with cis-DDP (2), with biplatinum compounds, bi-Pt (3), in complexes DNA + cis-DDP + bi-Pt (4) and DNA + bi-Pt + cis-DDP (5). The absorption spectra in complexes with Pt1 (**A**) and normalized absorption spectra in complexes with **Pt1** (**B**), **Pt2** (**C**), **Pt3** (**D**) and CD spectra of DNA in the test experiment for the competition between cis-DDP and **Pt4** for the binding site on a DNA molecule in a 0.005 M NaCl: (**C**) (bi-Pt) = 3 × 10^−5^ M. Explanations in the text.

**Figure 10 polymers-14-02044-f010:**
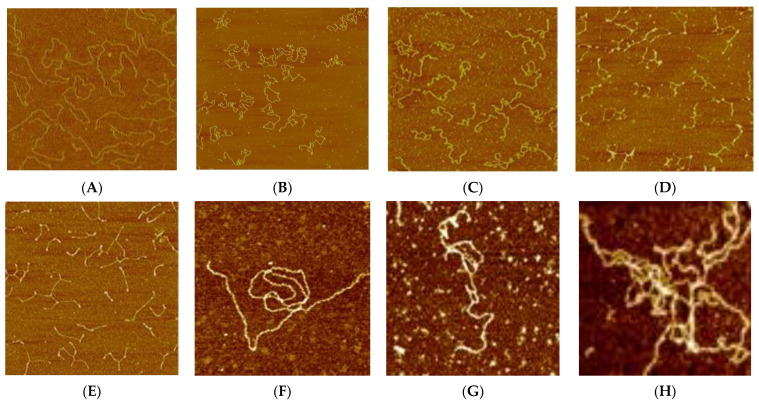
AFM images of DNA (**A**) and DNA complexes with cis-DDP (**B**,**C**), trans-DDP (**D**,**E**), **Pt1** (**F**), **Pt2** (**G**), and **Pt3** (**H**). Size of images is 2 μm (**A**), 3 μm (**B**–**E**) and 1 μm (**F**–**H**), (**B**,**D**) Fixation on mica was carried out at C(DNA) = 0.001 %, C(cis-DDP) = 4 × 10^−5^ M (**B**) and 5.5 × 10^−5^ M (**C**) C(trans-DDP) = 3.9 × 10^−5^ M (**D**) and 5.3 × 10^−5^ M (**E**). C(Pt1) = C(Pt2) = C(Pt3) = 5 × 10^−5^ M. C(MgCl_2_) = 5 × 10^−4^ M.

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
