# Peer review of "DNA Conformational Changes Induced by Its Interaction with Binuclear Platinum Complexes in Solution Indicate the Molecular Mechanism of Platinum Binding"

_polymers, 2022, doi:10.3390/polym14102044_

Round 1

Reviewer 1 Report

In this contribution Kasyanenko and coworker described the binding mode of various platinum complexes with DNA using a range of methods: circular dichroism, luminescent spectroscopy, low gradient viscometry , flow birefringence and AFM. This is in my opinion a nice piece of work and the manuscript is globally well written. I therefore recommend the publication of this work after correction of few points listed below:

1- In the title of the manuscript: binuclear platinum complexes instead of binuclear platinum compounds. Please correct this point in various other places in the manuscript.

2- When authors talk about competition between platinum complexes and fluorescence dye DAPI: does it correspond to the characterization technique called FID (Fluorescent Intercalation Displacement) assays ? If so, this term should be used in the manuscript.

3- The FRET-melting method is also generally used for characterization of interraction of Pt complexes with DNA: It should be mentioned in the manuscript and it should be explaned why the authors did not used this technique.

4- Pt complexes have often affinity for G-quadruplex DNA (see eg the work of Teulade Fichou on this field eg  Molecules 2019, 24, 404; Chem. Eur. J. 2015, 21, 7798; Org. Biomol. Chem. 2009, 7, 2864), it should be mentioned in the manuscript. Taking into account the involvement of N7 guanine in binding of this Pt complexes, an affinity for G-quadruplexes is plausible: did the authors consider this point (I do not ask extra extra experiment here, this is just a curiosity)

5- Complex numbers/names should be systematically in bold within the manuscript.

6- Figure 1 in unclear: this provide the full structure of A-C with the tetrazole fragment included. the charge of the complex should be indicated at the top right of the bracket (+ or 2+) and Cl or Cl2 should be replaced by Cl- and 2 Cl-. Why the complexes are called A-G  but also Pt1-Pt5. This is particulary confusing. This point must be solved before publication  

7- l19/l111 methyltetrazole

Author Response

Dear Reviewer!

Thank you very much for your careful reading and deep analysis of our article and for your valuable comments.

Below are the responses and comments on the comments made.

1- In the title of the manuscript: binuclear platinum complexes instead of binuclear platinum compounds. Please correct this point in various other places in the manuscript.

Author’s note:

According to this note we change the title of the manuscript:

DNA conformational changes induced by its interaction with binuclear platinum complexes in solution indicate the molecular mechanism of platinum binding

2- When authors talk about competition between platinum complexes and fluorescence dye DAPI: does it correspond to the characterization technique called FID (Fluorescent Intercalation Displacement) assays ? If so, this term should be used in the manuscript.

Author’s note:

Indeed, we have used the fluorescence dye DAPI and analyze its possible competition with platinum complexes for the binding sites in a minor groove. It was done to clarify the position of platinum complexes on DNA.  It should be noted that DAPI is not a classic intercalator, in contrast to EtBe, for example. However, the method used in this work is based on a similar approach. Therefore, we mentioned such a method in the text (see  line 92 and line 396 in the new variant of manuscript):

“ Fluorescent intercalator displacement (FID) is one of the methods used to analyze the binding of   ligands to DNA [[36, 42, 43].  ].  It is a convenient tool for understanding the type of binding and for assessing the relative binding affinities of compounds to DNA. A dye molecule with a greater fluorescence intensity when bound to DNA is used. It can be displaced by a ligand during its binding to DNA. As a result, one can see a reduction in fluorescence intensity of dye. The traditional FID is based on the intercalation of the dye molecule into DNA. Nevertheless, any mode of dye binding to DNA which may actually causes a decrease in fluorescence after dye release from DNA to solution  can be used.”

  1. Morel, E.; Beauvineau, C.; Naud-Martin, D.; Landras-Guetta, C.; Verga, D.; Ghosh, D.; Achelle, S.; Mahuteau-Betzer, F.; Bombard, S.; Teulade-Fichou, M-P. Selectivity of Terpyridine Platinum Anticancer Drugs for G-quadruplex DNA. Molecules 2019, 24, 404;
  2. Boger, D. L.; Fink, B. E.; Brunette, S. R.; Tse, W. C.; Hedrick, M. P. A simple, high-resolution method for establishing DNA binding affinity and sequence selectivity. J. Am. Chem. Soc. 2001, 123, 5878–5891.
  3. Lewis MA, Long EC. Fluorescent intercalator displacement analyses of DNA binding by the peptide-derived natural products netropsin, actinomycin, and bleomycin. Bioorg. Med. Chem. 2006;14:3481–3490.

“Fluorescent intercalator displacement (FID) is a convenient tool for understanding the type of binding and for assessing the relative binding affinities of compounds to DNA. We have used the fluorescence dye DAPI and analyze its possible competition with platinum complexes for the binding sites in a minor groove of DNA. It was done to clarify the position of platinum complexes on DNA.  It should be noted that DAPI is not a clas-sic intercalator, in contrast to EtBe, for example. However, the method used in this work is based on a similar approach.”

3- The FRET-melting method is also generally used for characterization of interaction of Pt complexes with DNA: It should be mentioned in the manuscript and it should be explained why the authors did not used this technique.

Author’s note:

We insert new text into manuscript (see line 77 in the new version):

“We should also mention such methods of analysis of DNA interaction with platinum compounds as FRET-melting method  and Fluorescent intercalator displacement (FID) assay.FRET-melting method is one of the effective approach for the analysis of the interaction of nucleic acids with different ligands, including  platinum complexes with DNA quadruplexes [36].  It is known that platinum complexes have affinity for G-quadruplex DNA [37, 38]. G-quadruplexes formed in human telomeres are considered attractive targets for anticancer drugs. For example, it was shown that telomeres in cisplatin-treated HeLa cells are markedly shortened and degraded [39]. Possibly, platinum affinity for G-quadruplex DNA and the role of N7 guanine in binding of Pt complexes to DNA may explain the activity of cisplatin.

The fluorescence-based Förster Resonance Energy Transfer-melting method is based on the stabilization of nucleic acid structure induced by ligands. This method has been used, for example,  to estimate whether a compound is a good quadruplex ligand or not [40, 41]. We did not use this method in our work due to the specific requirements for the ligands used and DNA modification, which is difficult in the case of high molecular weight samples.”

  1. Morel, E.; Beauvineau, C.; Naud-Martin, D.; Landras-Guetta, C.; Verga, D.; Ghosh, D.; Achelle, S.; Mahuteau-Betzer, F.; Bombard, S.; Teulade-Fichou, M-P. Selectivity of Terpyridine Platinum Anticancer Drugs for G-quadruplex DNA. Molecules 2019, 24, 404;
  2. Trajkovski, M.; Morel, E.; Hamon, F.; Bombard, S.; Teulade-Fichou, M-P.; Plavec, J. Interactions of Pt-ttpy with G-Quadruplexes Originating from Promoter Region of the c-myc Gene Deciphered by NMR and Gel Electrophoresis Analysis. Chem. Eur. J. 2015, 21, 7798-7807
  3. Bertrand, H.; Bombard, S.; Monchaud, D.; Talbot, E.; Guédin, A.;   Mergny, J-L.;  Grünert, R.;   Bednarskid, P. J.;  Teulade-Fichou, M-P.  Exclusive platination of loop adenines in the human telomeric G-quadruplex. Org. Biomol. Chem., 2009, 7, 2864-2871
  4. Ishibashi, T.; Lippard, S. J. Telomere loss in cells treated with cisplatin. Proc Natl Acad Sci USA. 1998, 95, 4219–4223).
  5. Renciuk, D.; Zhou, J.; Beaurepaire, L.; Guedin, A.; Bourdoncle, A.; Mergny, J. L., A FRET-based screening assay for nucleic acid ligands. Methods 2012, 57, 122-128
  6. Luo, Y.; Granzhan, A.; Verga, D.; Mergny, J.-L. FRET-MC: a fluorescence melting competition assay for studying G4 structures in vitro. Biopolymers, Wiley, 2021, 112, e23415.

4- Pt complexes have often affinity for G-quadruplex DNA (see eg the work of Teulade Fichou on this field eg  Molecules 2019, 24, 404; Chem. Eur. J. 2015, 21, 7798; Org. Biomol. Chem. 2009, 7, 2864), it should be mentioned in the manuscript. Taking into account the involvement of N7 guanine in binding of this Pt complexes, an affinity for G-quadruplexes is plausible: did the authors consider this point (I do not ask extra extra experiment here, this is just a curiosity)

Author’s note:

Thank you very match for this remark. G-quadruplexes formed in human telomeres are considered attractive targets for anticancer drugs. Indeed, platinum affinity for G-quadruplex DNA may explain the activity of cisplatin. It was shown that telomeres in cisplatin-treated HeLa cells are markedly shortened and degraded (T. Ishibashi and S. J. Lippard. Telomere loss in cells treated with cisplatin. Proc Natl Acad Sci USA. 1998 95(8), 4219–4223).

We have inserted the following text into the manuscript (see line  81 in the new version of manuscript):

“It is known that platinum complexes have affinity for G-quadruplex DNA [37, 38]. G-quadruplexes formed in human telomeres are considered attractive targets for anticancer drugs. For example, it was shown that telomeres in cisplatin-treated HeLa cells are markedly shortened and degraded [39]. Possibly, platinum affinity for G-quadruplex DNA and the role of N7 guanine in binding of Pt complexes to DNA may explain the activity of cisplatin.”

  1. Trajkovski, M.; Morel, E.; Hamon, F.; Bombard, S.; Teulade-Fichou, M-P.; Plavec, J. Interactions of Pt-ttpy with G-Quadruplexes Originating from Promoter Region of the c-myc Gene Deciphered by NMR and Gel Electrophoresis Analysis. Chem. Eur. J. 2015, 21, 7798-7807
  2. Bertrand, H.; Bombard, S.; Monchaud, D.; Talbot, E.; Guédin, A.;   Mergny, J-L.;  Grünert, R.;   Bednarskid, P. J.;  Teulade-Fichou, M-P.  Exclusive platination of loop adenines in the human telomeric G-quadruplex. Org. Biomol. Chem., 2009, 7, 2864-2871
  3. Ishibashi, T.; Lippard, S. J. Telomere loss in cells treated with cisplatin. Proc Natl Acad Sci USA. 1998, 95, 4219–4223).

5- Complex numbers/names should be systematically in bold within the manuscript.

Corrected

6- Figure 1 in unclear: this provide the full structure of A-C with the tetrazole fragment included. the charge of the complex should be indicated at the top right of the bracket (+ or 2+) and Cl or Cl2 should be replaced by Cl- and 2 Cl-. Why the complexes are called A-G  but also Pt1-Pt5. This is particulary confusing. This point must be solved before publication

Corrected

7- l19/l111 methyltetrazole

Сorrected

Authors

Reviewer 2 Report

The results reported in the manuscript are concerned to the interactions of DNA molecules with binuclear platinum complexes. In general the results are looking good and the conclusions are well proved by experimental material. The minor question arises:

Studying the electronic spectra of DNA adducts authors assumes that the absorption of free binuclear compounds is the same as their absorption in complexes with DNA. Is that means that only electrostatic interactions are supposed?

Author Response

Dear Reviewer!

Thank you very much for the reading and analysis of our article and for appreciating our research.

In response to your question, we would like to point out the following

Question:

“Studying the electronic spectra of DNA adducts authors assumes that the absorption of free binuclear compounds is the same as their absorption in complexes with DNA. Is that means that only electrostatic interactions are supposed?”

Answer to question:

Indeed, the electronic spectra of free binuclear compounds and their absorption in complexes with DNA are similar. We can conclude that the chromophores of platinum compounds under study do not participate in direct contact with DNA molecule. This conclusion concerns only chromophores that absorb in the wavelength range used. In this case, both the coordination of platinum to the atomic groups of DNA and the electrostatic interaction can occur.

Thanks again for your feedback and valuable questions.

    Sincerely,

    Authors
